# State-of-the-Art Methods to Improve Energy Efficiency of Ships

**Johannes Hüffmeier** [1,*] **and Mathias Johanson** [2]

1    Material & Production Division, RISE Research Institutes of Sweden, 413 46 Gothenburg, Sweden
2    R&D Department, Alkit Communications, 431 37 Mölndal, Sweden; mathias@alkit.se
*    Correspondence: johannes.huffmeier@ri.se; Tel.: +46-70-580-62-44

**Abstract:** Generating energy efficiency through behavioural change requires not only understanding and empathy with user interests and needs but also the fostering of energy saving awareness, a technique and framework that supports operators and ship owners. There is strong potential to make use of different technical solutions to increase energy efficiency, but many cost-efficient solutions relate to carrot-and-stick incentives for operators to minimise energy consumption. These incentives range from voyage planning with weather routing eco-driving bonus, to torque limitations and changes in company policies, all of which demonstrate that the operators' on-board importance for the energy consumption has been identified. Data collection will allow operators to make better decisions in the lifecycle of the ship from knowledge-driven design to operation, redesign and lifetime extension. Various systems are available for data acquisition, storage and analysis, some of which are delivered by well-known marine suppliers while others are stand-alone systems. The lack of standardization for data capture, transmission and analysis is a challenge, so systematic improvement is required in shipping companies to achieve energy savings. When these are achieved, they will be the result of customer requirements, cost pressure or individual driving forces in the companies. The potential energy savings, brought up in interviews, shows up to 35% on specific routes and up to 60% in specific maneuvers. These savings will be made feasible by operators and crews being involved in the decision-making process.

**Keywords:** energy efficiency; climate change; maritime digitalization; IoT; machine learning; user involvement; data-driven design; policy making; green shipping; improved operations



## 1. Introduction

The EU has set strict targets for the use of renewable energy, and several countries have set up plans and sub-goals to meet them. While target fulfilment of the electricity and heating targets is within reach or has been surpassed, the transport sector in almost all European countries is far behind these targets. Because this sector accounts for a large part of total energy use and climate impact, it is of great importance to find different ways to reduce its environmental impact, especially where shipping is concerned. Seaborne trade accounts for 78% of global trade in metric tonnes, and of that liquid and dry bulk comprises 84%. [1] Swedish shipping accounts for between 70 and 90% of all imports and exports that go in and out of Swedish ports. Shipping accounts for about 4% of the total domestic transport sector's greenhouse gas emissions in Sweden, and if the infrastructure and international transports are included, shipping and its ports and routes amount to about 30% [2]. Globally, shipping contributes about 2.4% of global greenhouse gases [3] from international trade. If nothing is done, these emissions will continue to grow as the sector grows, driven by a larger population and demands for higher prosperity. Demand for certain bulk products and container cargo is expected to grow [4] along with an estimated 50–250% increase in $CO_2$ emissions from shipping from 2012 to 2050 [3].

Improved energy efficiency is needed on vessels, which often does not fall under legal requirements to reduce emissions and energy use [5]. To make a difference in the climate impact of shipping and to reduce energy consumption onboard ships, it is not enough

to collect high level-data such as MRV (Monitoring, Reporting, Verification) and work with spreading and calculations methods; more detailed data and structured methods are needed. The statistics derived from MRV, for instance, do not seem to be specific enough to make competent policies on how to reduce GHG (greenhouse gas) emissions or on how ship owners should increase energy efficiency. Therefore, more detailed data collection and analysis methods are needed to make shipping more energy efficient [2].

Most often, the contract between the transport buyer or cargo owner and the shipowner and the low cost of fuel have not driven reduced energy consumption. Some comprehensive studies have been performed on the energy efficiency of larger cargo vessels and ferries [6,7]. The causes and effects are often more complex compared to those of smaller vessels since a larger ship's final energy consumption depends upon many systems that can affect each other. If one researches smaller vessels, it is usually easier to understand the connections between energy conversion, consumption and use as their movements take place in the same geographical areas or as they run on the same routes. The knowledge learned here can later be scaled up to larger vessels. The smaller vessels sail close to the coast and thus have a greater impact on air quality in densely populated areas. In addition, they usually operate on smaller margins and shorter-term contracts where fuel costs have a greater economic impact, which means that energy efficiency also represents a clear economic incitement. Often, however, there is no detailed evidence of how energy consumption occurs. Measurement data are available on distances travelled and fuel consumption, but not enough is known about which factors have the greatest impact on energy consumption. The analysis should be based on real operating data from ships, where previous research has shown that there is a significant difference between how a ship works and how it was originally designed to be used. The importance of starting from realistic measurement data and the need for advanced analysis methods is shown, among other things, by Ahlgren [6]. The data analysis of energy consumption is often complex and there are different driving forces that go into decision-making. However, increased data collection can be unprofitable if there are no methods to analyse the complex systems. Developments in machine learning provide new opportunities to develop both technically and economically powerful tools for energy efficiency.

Worldwide, 90% of all data has been created in the last 2 years. This correlation has been known for quite some time, but a large part of the new knowledge that has been collected is rarely analysed [8,9]. The work of this paper is part of a project that aims to collect and analyse data from a range of smaller vessels to derive tools and services for achieving higher energy efficiency. The vessels will be analysed in detail to reduce their energy impact by 10–35% thereby giving decision support to the crews and onshore personnel. Making use of smaller vessels with a limited number of systems makes it easier to identify the drivers of high energy consumption since it reduces the complexity of deriving a useful correlation between internal and external factors and the energy. It is also easier to ensure the relevance of the results for similar vessels.

The data analysis of energy consumption is often complex and there are different driving forces for decisions. However, increased data collection can be unprofitable if there are no proper methods to analyse the complex systems. Developments in machine learning provide new opportunities to develop both technically and economically powerful tools for energy efficiency. Online and offline data analyses that relate to various statistical distributions can lead to the identification of optimal operation conditions of vessels [10]. Even today, to some extent, economic driving is applied, for example eco-driving. However, the effect on energy consumption is limited in many cases as energy efficient decision-making is too complex for the operator/navigator without access to real time data. In addition, there is not always an incentive for individuals to reduce energy use [11]. Data collection is increasing, but both quality review and analysis are not carried out to the same extent.

A telling example of the importance of knowing and being able to analyse data for a successful energy efficiency comes from two Swedish bunker boats. Both vessels met the requirements specification, but the feeling of the crew that knew both vessels was that the last one delivered was not as powerful and did not come up to the same speed as the first.

Through careful analysis it was found that the efficiency of the propeller on the other vessel was very low. By optimizing the propeller, the vessel became 1 knot faster and reduced energy consumption by 25%, the most effective measure the company has undertaken [12].

In this case, access to relevant data together with the crew's experience showed a great improvement potential for energy efficiency. The cause could be identified, and an improvement implemented. Through new technology and new analytical methods for complex systems linked to machine learning, effective support tools can be developed that facilitate on-board decision making. The development of strategies for a ship's energy efficiency and the efficient use of energy implemented as a management strategy has a direct effect on the energy consumption per commodity/ton/volume carried [13]. Sustainable energy management processes must be considered together with power management strategies in ship operation where the power management of vessels is fuel-bound and depends on external factors, technical efficiencies and the efficiency of operation maintenance processes [14]. The monitoring of energy consumption and ship performance has traditionally focused on the vessel condition, particularly hull and propeller fouling [15]. The implementation of energy management systems today implies optimization of the power management of the engine and propulsors, but it also increasingly considers load handling and HVAC (Heating, Ventilation and Air Conditioning) systems [7].

Reduced energy consumption is only possible when motivating and involving the crews and operators to save energy. Decisions need to be based on a systematic approach to data collection (described in its different stages in this paper) that identifies potential savings, provides real-time decision support and allows to follow up on long-term effects.

## 2. Materials and Methods

*Methods Used*

To derive continuous improvements in decreased $CO_2$ emissions and increased energy efficiency, a systematic approach is needed that involves management as well as crews, as shown in Figure 1. Most important is the development of a management strategy that will develop these improvements in the ship's operation [16]. This paper describes a quality management-based approach and focuses on the parts in the upper part of the cycle shown in Figure 1.

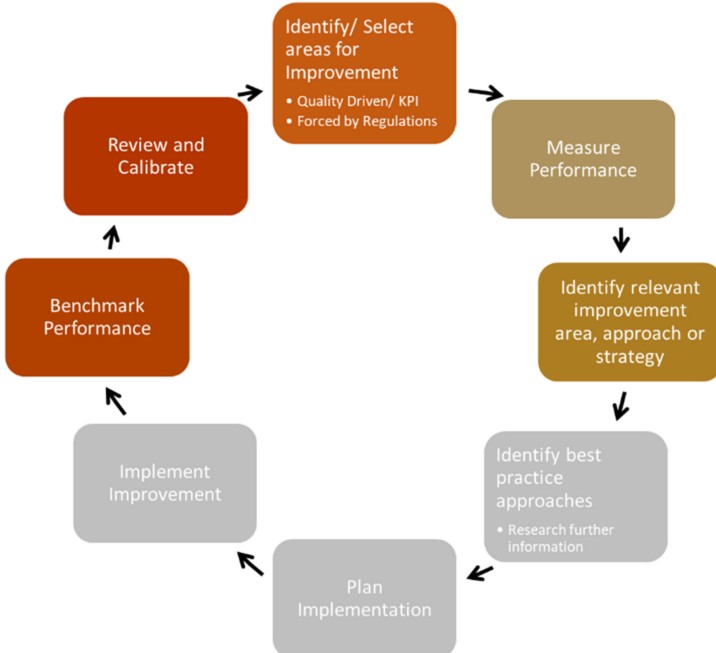

**Figure 1.** The Improvement Cycle for decreased $CO_2$ Emissions and increased Energy Efficiency. This paper focuses on the upper half of the cycle.

This work builds upon earlier research experiences: a study in the literature; interviews with ship owners, crews and experts; and on structured Internet searches. The interviews were semi-structured with open-ended questions. Because the informants had different roles and different organizational placements, the interview questions were adapted each time: some were conducted on site while others were conducted by telephone or online. The literature study focused on energy efficiency, human performance and decision support in a shipping context. Research on the role of humans in energy efficiency is limited. There are traces in the literature of research into human decision-making, data collection, and HTO (Human-Technology-Organisation) in general decision-making for energy savings, but very little seems to have been translated into applied projects or maritime studies. A dominant focus for the human role in fire protection at sea is evacuation. Therefore, the search was extended to areas that did not directly affect the maritime industry, but where parallels could be made for shipping and the design of technical installations. The searches were also supplemented with references from previously completed projects, including for the area of technical solutions. The empirical data were coded in batches, and a thematic analysis generated different themes. The analysis was carried out iteratively where the data were examined in relation to theory. After the empirical data were satisfactorily sorted they were converted into a running text in the report, based on the identified themes and the derived improvement cycle.

Little has been published in relevant journals on these methods and experiences, so our focus was on expert consultations and publicly available information as well as on public and project internal data sources and measurements to highlight the different stages of the improvement cycle. The most important prerequisite for improving energy efficiency is the analysis of energy data in order to improve a ship's system and its operation. The overarching goal of the analysis of energy data is to extract the most information out of the available data [17]. Decision support tools have been derived for various applications in the maritime industry [18]. The implementation of a successful decision support system has prerequisites such as simplicity, comprehensibility and reproducibility [19]. To have an impact, the decision support system must meet the needs of the user, be trusted, and produce results and findings that are concise [20].

## 3. Identification and Selection of Areas for Improvement

### 3.1. Quality Driven Selection—Decarbonisation of Transport and Ships

Decarbonization can be achieved by a number of, and possibly a combination of, different approaches:

- Reducing the demand for freight transport,
- Reducing the carbon content of freight transport energy,
- Shifting freight to lower carbon transport modes,
- Optimizing vessel loading, and
- Increasing the energy efficiency of freight movement

None of these decarbonization efforts is achievable only by technological or operational advancements: all require policies, cost reductions, or customer demands. For a ship owner who works systematically with quality, risk and environmental management, energy efficiency improvements will be a part of the day-to-day work. Interviewees showed that crews on vessels set weekly and daily targets that they could follow up together with the onshore personnel. This implies that the targets, adjusted for weather or schedule requirements, lead personnel to focus on energy consumption and efficient operations [21]. The results from [22] showed that companies "that have [a] majority of their ships on time charter have a higher implementation of energy efficiency technologies compared to firms that operate ships on the spot charter." This implies that a regular customer can directly or indirectly set requirements on more efficient ships and ship operations even though the principal (charterer/shipper) pays for the fuel but is not responsible for energy efficiency investments.

### *3.2. Rule-Based Selection and Identification*

#### 3.2.1. Rules and Regulations—International

Shipping and aviation are seen as the modes of transportation where reducing carbon emissions is the most difficult [23]. The International Maritime Organization (IMO) has for some time been working on new regulations and targets relating to the carbon footprint of shipping. Reports published by the IMO estimates that carbon dioxide emissions from shipping were equal to 2.8% and 2.2% of global human-made emissions in 2007 and 2012, respectively. It is expected that these could rise by 50% to 250% by 2050 if no action is taken [24]. One of the main barriers to the effective implementation of energy efficiency measures is the heavy reliance it places on flag states for inspection of vessels and enforcement of provisions. Some of these states are not signatories to all Maritime Pollution Prevention (MARPOL) annexes, which also hampers implementation [25].

#### 3.2.2. IMO GHG Target

Therefore, IMO members have agreed to set targets for the reduction of $CO_2$ emissions per transport work, as an average across international shipping, by at least 40% by 2030 and to pursue efforts towards 70% by 2050, compared to 2008 emissions. The updated numbers from 2020 [26] indicate that ship emissions are projected to increase from about 90% of 2008 emissions in 2018 to 90–130% of 2008 emissions by 2050 "for a range of plausible long-term economic and energy scenarios". The COVID-19 pandemic is also expected to result in a small but demonstrable reduction in emissions over the next few years.

Based on the ICCT [27] report and the IMO report, the development of carbon emissions shown in Figure 2 can be derived.

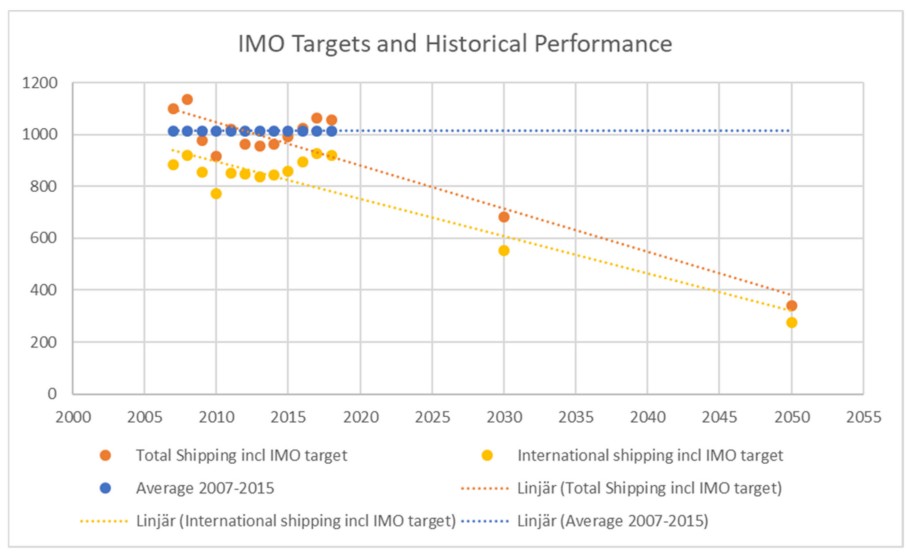

**Figure 2.** IMO targets and historical performance towards based on IMO and IPCC data.

#### 3.2.3. IMO EEDI and SEEMP

Since 2011 the "International Convention for the Prevention of Pollution from Ships (MARPOL 73/78)" has included a new Chapter 4 containing Regulations on Energy Efficiency for ships in the Annex VI; specifically, the "Energy Efficiency Design Index (EEDI) for new ships", and "the Ship Energy Efficiency Management Plan (SEEMP) for all ships" are important. It includes new definitions, requirements for surveys, and validates the formats for "the International Energy Efficiency (IEE) Certificate". Furthermore, the role of the energy efficiency management of new ships is emphasised as the later ship's operation through represents a major contributes. The IMO legislation on the Energy Efficiency Design Index, or EEDI, is currently moving into its fourth and last phase to increase energy efficiency on new buildings by up to 30% in 2025 based on 2013 data. The latest official evaluations [26] show that most ship types, despite large container vessels, have just improved

efficiency by up to 3%. The drivers towards reduced emissions and increased efficiency are therefore to be found in operational measures including ship speed reduction.

3.2.4. MRV—Monitoring, Reporting and Verification of $CO_2$ Emissions (EU + IMO)

The IMO and EU have in parallel established rules for collecting data and calculating ship efficiency. Both aim for similar follow-ups and plans regarding emission reduction, but have different focuses and requirements:

EU MRV—EU Monitoring, Reporting and Verification of $CO_2$ emissions.

Data collection started on 1 January 2018, and includes Ttotal transport work, time at sea and in port, cargo carried, and average energy efficiency. All ships calling in at EU ports are obliged to follow these regulations. The main obligations for companies eligible under the EU MRV Regulation are

(1) Monitoring: From 1 January 2018, companies shall—in line with their respective monitoring plans—monitor for each of their ships $CO_2$ emissions, fuel consumption and other parameters, such as distance travelled, time at sea, and cargo carried on a per voyage basis, so as to gather annual data into an emissions report submitted to an accredited MRV shipping verifier.

(2) Emissions report: From 2019, by April 30 of each year, companies shall submit to the Commission and to the States in which those ships are registered ("flag states") a satisfactorily verified emissions report for each ship that has performed maritime transport activities in the European Economic Area during the previous reporting period (calendar year).

(3) Document of compliance: From 2019, by June 30 of each year, companies shall ensure that all their ships that performed activities in the previous reporting period and are visiting ports in the European Economic Area carry on board a document of compliance issued by Thetis MRV. This obligation might be subject to inspections by Member States' authorities.

- IMO DCS—IMO Data Collection System on fuel consumption (data collection started 1 January 2019).
- Ship Energy Efficiency Plan (SEEMP): establishes a mechanism for ship owners to improve the energy efficiency of both new and existing ships

3.2.5. Rules and Regulations—National

National regulations can govern vessels that are non-SOLAS (International Convention for the Safety of Life at Sea) and bear the flag of the specific country. Legislation on emissions, such as is done regarding Norwegian fjords is another example on how national rules can govern "international" ships.

Domestic vessels were underestimated in the previous calculations that IMO performed. The new estimate made in 2020 [26], implies that about 30% of the world's fleet falls under national legislation and governmental responsibility. As an example, it describes Sweden's ambitious target of becoming fossil-fuel free by 2045, and IMO aims to minimise emissions by 50% by 2050. While there is ongoing development in road transport, other parts of the transport sector progress more slowly. Increased energy efficiency has a huge potential, and here the maritime sector lags behind many other industries. This national target does not imply a direct change, but based on increased customer demands, which can be a procurement processes by the state, regions or counties, it has a direct influence on environmental performance [7].

**4. Measure Performance**

*4.1. High Level Data Collection—MRV*

An analysis of the first set of reported data for the EU MRV regulations revealed huge differences in fuel consumption for the vessels, despite third-party verification of the reported data. The European Maritime Safety Agency (EMSA) has now released the 2018 MRV results for the fleet affected by the new regulations. While just looking at the

numbers as such, one can observe that there is no standard on what has been collected, which adds a high level of uncertainty. As shown in Figure 3, the different verification entities have used very different measures to compare fuel consumption. While there are certainly differences among ship types, ship sizes and trading areas, there should not be such a divergence as indicated in the database. When collecting data, EMSA should probably put more effort into standardizing what is reported and how. The following disclaimer is therefore not sufficient to support trustworthy reporting on ship emissions in Europe: "All emission data is entered by Companies and confirmed by Verifiers accredited by EU Member States National Accreditation Bodies. The European Commission and EMSA decline any responsibility or liability whatsoever for errors, deficiencies in these data or its accuracy".

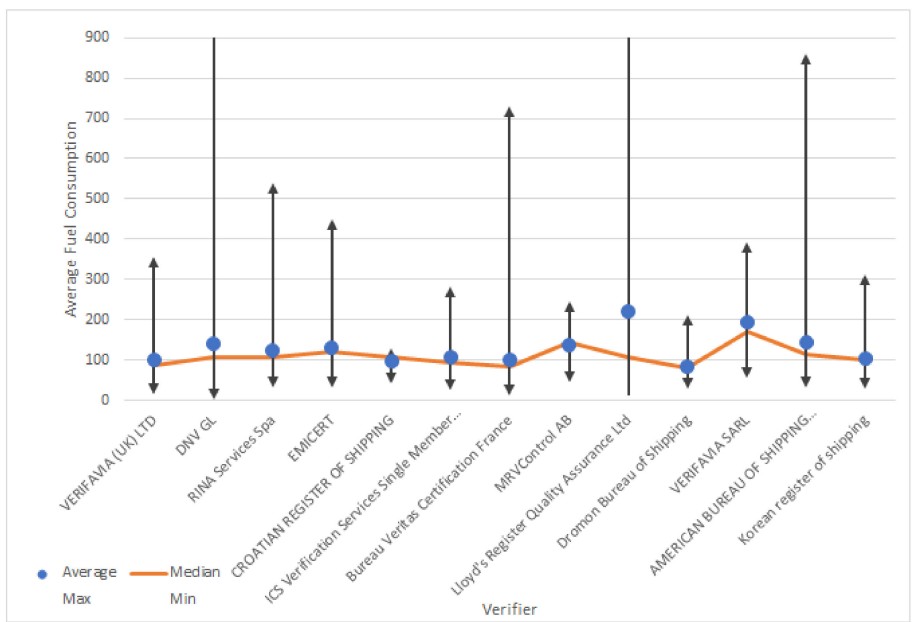

**Figure 3.** Differences in fuel consumption data (kg/nm) grouped by the different third-party verifiers shown by average (blue), median (orange), maximum and minimum values (arrow).

From looking at four sister vessels (tankers), which should have an average fuel consumption on the same order of magnitude, the following can be found. While three vessels were verified by one company, the other was validated by another entity. A difference of almost 20% in average fuel consumption was found.

An example of a container vessel fleet of 5 vessels, verified by the same company, showed differences of 15% from the most deviating vessel to its sister vessels.

The biggest differences found in this analysis related to a fleet of bulk carriers. Five different verifiers looked into the vessels' performance, resulting in an annual average fuel consumption per distance (kg/n mile) of 111 but with a standard deviation of more than 23, the figure varied significantly. As this might partly be explained by the difference in loaded and unloaded voyages, this is not reflected in the statistics on consumption (Annual average fuel consumption per transport work (mass) (g/m tonnes·n miles)), where these differences are even higher.

To make a difference when it comes to the climate impact on shipping and reduction in the energy consumption onboard ship, it is not enough to collect high level data and work with spreading and calculations methods; more details are needed.

### 4.2. Detailed Data Collection—Energy Management Systems

To increase the energy efficiency of ships, knowledge of specific vessels and routes needs to be established, and data need to be acquired on the main consumers with an

adequate time resolution. Figure 4 shows the user interface of the Blueflow Energy Management System, where the main parameters for the trips of a ferry are summarised. Based on interviews, such systems allow the crew and responsible people onshore to get a detailed overview for each trip and provide decision support. As each trip differs from the other, the operator can learn how consumption correlates to different decisions and start to optimise route planning and execution. In the shipping company, the crew and onshore personnel can discuss measures of improvement. The interviewees indicated that changes to the schedule, technical improvements and a change in company internal policies had a direct impact in the range of 20–35%, for certain maneuvers even as much as 60%. At the same time, changes to some equipment could have had a negative impact, where a ship owner changed the main engine to a newer and more fuel-efficient model, but with a higher-rated power. As the operators had a common way of running the vessel based on its steering devices, energy consumption increased instead of decreasing, as the engine was run most of the time at full ahead, which increased consumption as well as speed.

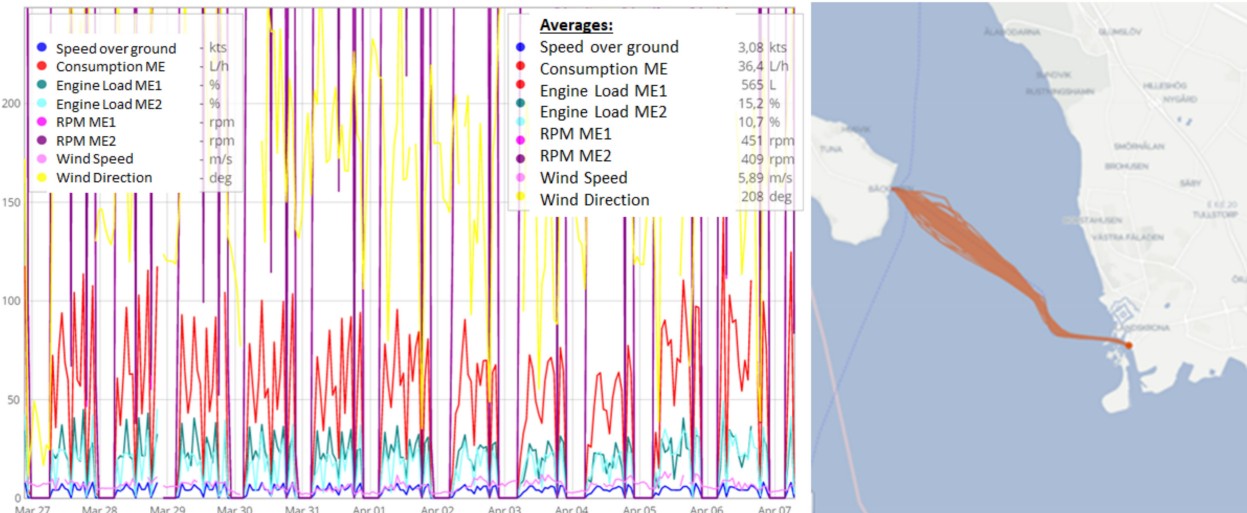

**Figure 4.** Screen shot of the state-of-the-art Energy Management System from Blueflow for the vessel Uraniborg sailing between Landskrona and the Island of Ven, Sweden.

## 5. Identify Relevant Improvement Area, Approach or Strategy

*Techniques for Energy Savings—Shipping towards Fossil-Fuel Free Transport*

Based on interviews and literature study, the collected data can be used to identify relevant improvement areas. There are various methods and technical solutions available to move shipping towards fossil-fuel free shipping. Some examples are given below.

- Biofuels, bio-oils and liquefied natural gas (LNG) are slowly being introduced into the shipping industry as possible replacements.
- LNG as an alternative fuel has been on the rise since the 2000s
- Electrification of local traffic (ferries, public transport) has been on the rise over the last five years, in parallel with even HVO (Hydrogenated Vegetable Oil) as a fuel
- Electrical drives for ships were around for a long time powered by fossil fuels, but a shift to batteries has started on new installations.
- Methanol as a fuel has been tested mainly in Sweden and on methanol carriers in a limited fleet so far.
- Tests and prototypes are built with ammoniac, "sails", Flettner rotors, and hydrogen.
- Biofuels mixed with traditional fuels have been implemented in various projects.

Increased energy efficiency is also needed despite the choice of alternative fuels. This increases the need for energy efficiency measures to be identified and implemented within the maritime sector. The EU-ordered report [28] has amongst others identified a couple of

measures to reduce GHG emissions from shipping. It is evident that all solutions relate to energy efficiency measures. These consists of speed reduction (10%), trim and ballast optimization, autopilot or power management adjustments, weather routing, speed control of pumps and fans, propeller polishing and hull cleaning, hull coating, propeller/rudder upgrade, main engine tuning and common rail, waste heat recovery, and energy saving utilities (e.g., lighting).

Adopting alternative fuels has been a slow process. The statistics, collected by DNV GL, indicate that only very few have invested and made use of cleaner alternatives, assuming a SOLAS fleet of 120,000–150,000 vessels [29] and smaller vessels following national regulations on the order of magnitude of 900,000 [30]. Many of the relevant vessels are owned and managed from Scandinavia. Vessels making use of HVO or biodiesel are not included in these statistics. Alternative fuels that allow for cleaner operations come partly with different drawbacks, such as price, taxes, availability, safety, and logistics. One central concern for the wider adoption of most of the fuels is the energy content of the different fuels for volume and weight as shown in Figure 5 below. As the payload of ships in many cases is determined by the available buoyancy and space onboard, volumetric energy density is a crucial factor. Very few of these alternatives are competitive in that respect to heavy fuel oil (HFO) and marine diesel/gas oil.

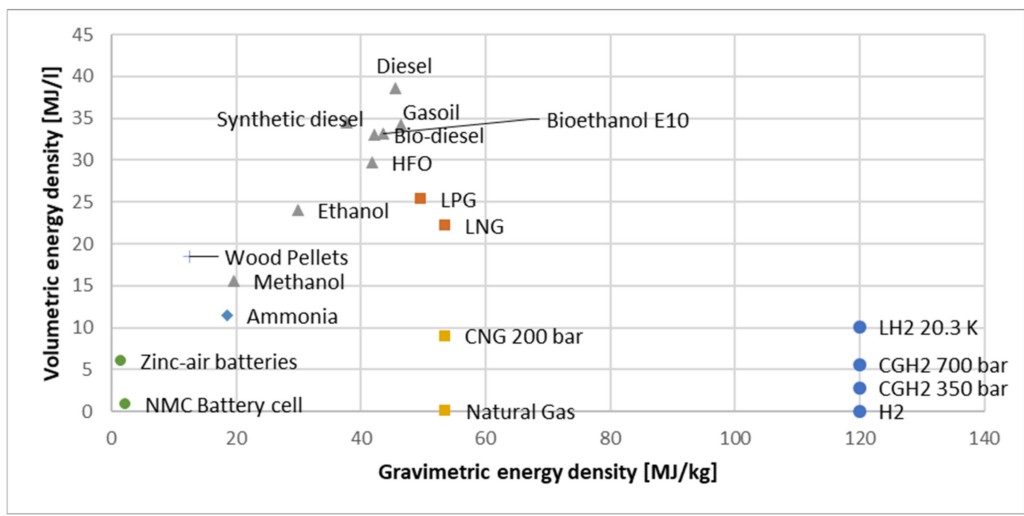

**Figure 5.** Energy Density of various fuels. For ships, volume is most often the critical parameter. Source for values: Wikipedia.

Regarding LNG as a solution, its critics have been around since the start. The ITTC study [31] has recently shown that LNG has limited effect on carbon emissions from shipping when looking at the lifecycle of the fuel from well to propeller. This is caused mainly by the high energy consumption of cooling LNG to its liquid state ($\sim-162$ °C). Therefore, LNG cannot be a solution from a climate-change perspective, while LBG (Liquefied Biogas) would make a difference.

The potential for the electrification of shipping is huge, especially for shorter distances. When it comes to longer distances, the energy content of batteries is too low to cover the whole journey. The potential of batteries is here more for peak shaving purposes. Battery systems can absorb load variations in the network so that engines only see the average and optimal system load. The system will then level the power seen by engines and offset the need to start new engines. Peak shaving will improve fuel efficiency and reduce engine running hours. Below the potential for different ship types is described. Operations of batteries allow for enhanced dynamic performance, spinning reserve, peak shaving, electrical enhanced ride through, strategic loading, and zero emissions operations.

Based on the above data, it can be concluded that use of alternative fuels as such requires increased energy efficiency to a large extent if one wants to maintain the reach of

the vessels. Battery storage can make a difference for short-distance shipping by making vessels totally electric, and with the increased energy efficiency could maintain their reach as shown in the Figure 6. For long-distance shipping, batteries could result in hybrid operations leading to a more efficient use of main and auxiliary engines, which would then result in partly more reliable operations, zero-emission modes and meeting possible customer or charter requirements. This has even been identified by other stakeholders such as Wallenius Marine [32].

| | | | |
|---|---|---|---|
| **Tugs:** potential of 20-30% of engine load can be run with batteries, reduced maintenance in the same order of magnitude, first concept with 100% battery operation | 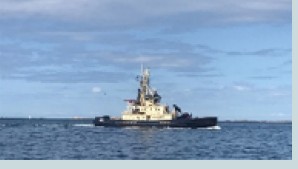 | **Cargo vessel:** Cargo handling can be evened out when operating cranes (30-40%). Reduced number of conventional auxiliary machines, reduced maintenance, electrification in ports | 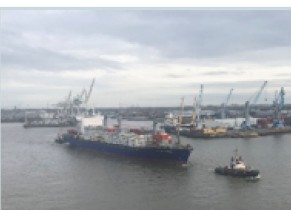 |
| **Ferry (smaller size, shorter trips):** Total electrification Significantly reduced maintenance (halving) | 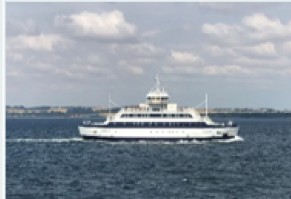 | **Ferry (larger and longer trips):** Combined operation with battery and standard driveline, optimized fuel consumption, battery operation during port stay and in zero emission areas | 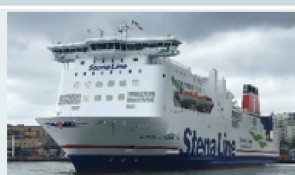 |
| **Smaller tankers** Pumps and hotel loads can be run via batteries, some tours can be run with batteries, the anchorage time can be run with battery operation | 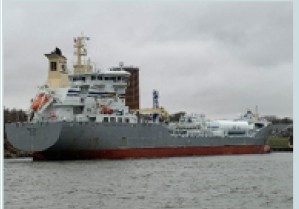 | **Larger Tankers** Potential savings in procurement due to reduced machine size Improved fuel economy and reduced maintenance | 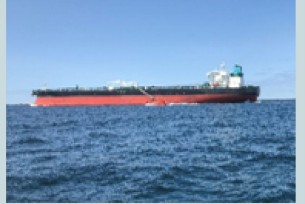 |
| **Work boats** Can be electrified to 100%, reduced maintenance, better operational capabilities, risk of shorter range | 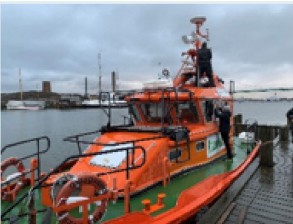 | **Ro-Ro ships** The ventilation can be run with the help of batteries in the port (30-40%). Reduced number of conventional auxiliary machines, reduced maintenance, electrification in ports and archipelago | 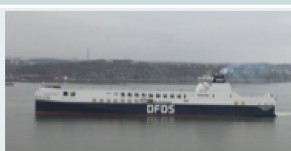 |

**Figure 6.** Potential of energy storage/battery usage in different applications.

## 6. Identify Best Practice Approaches

There are a few examples of how energy efficiency can be increased in the shipping sector. The best practices are described in this section.

### 6.1. Use of Wind Energy by Flettner Rotors or "Sails"

Making use of wind by sailing vessels has a huge potential. Instead of losses in traditional propulsion arrangements, wind can be directly used to move the vessel forward. Limitations are given by the speed that can be achieved by wind propulsion.

The Flettner rotor is based on an invention by Anton Flettner, based upon the Magnus effect. The system consists of a rotor—a smooth cylinder with disc end plates that are spun along its long axis. When air passes at right angles to it, the Magnus effect causes an aerodynamic force to be generated in the direction perpendicular to both the long axis and the direction of airflow as shown in the Figures 7 and 8 below.

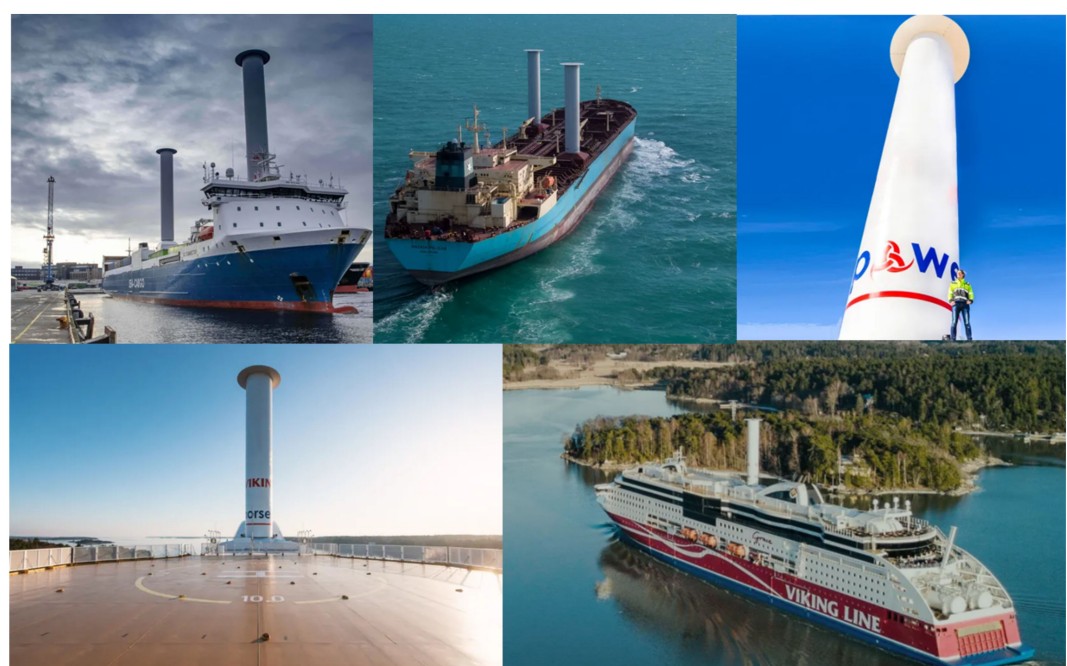

**Figure 7.** Examples of vessels with Flettner rotors installed, different vessel types: RoRo vessel, tanker and ferry, Image courtesy: Norsepower Oy Ltd.

After the "Buckau" prototype built by Flettner, only test vessels had been built until recently when the Flettner rotor got a revival. With *E-Ship 1*, a wind-turbine transport vessel launched in 2008, a new series of ships was equipped with these rotors. Energy savings of about 5–10% have been achieved with Flettner rotors, and savings of up to 20% are possible on favourable routes [33].

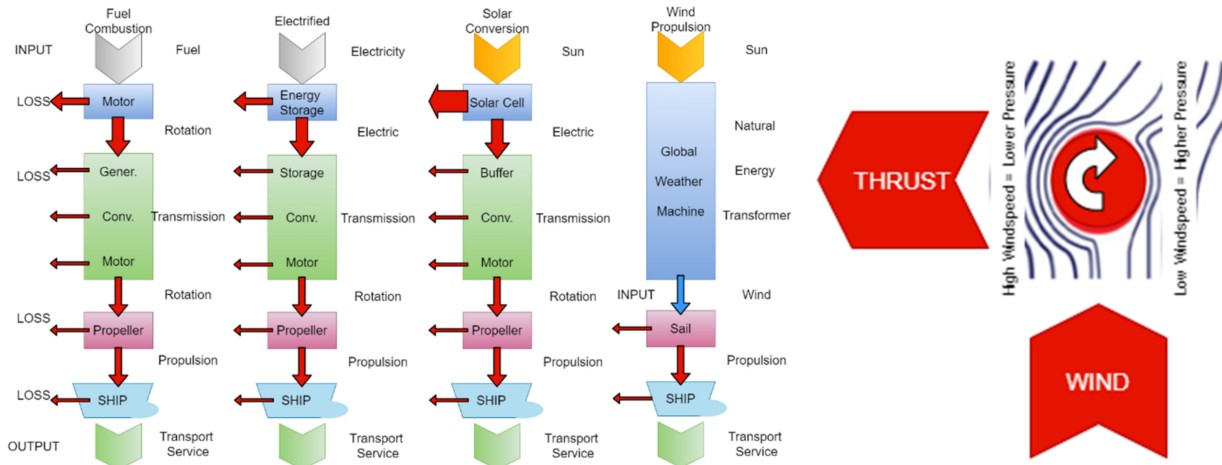

**Figure 8.** Modes of Energy Transformation for Ocean Transport and the Magnus Effect used by the Flettner rotors adapted from [34].

Concepts for a wind powered pure car carrier have been derived—as shown in Figure 9—that use upright standing wings, reach sailing speeds of around 10 knots which results in 12 days for a typical Atlantic crossing. The vessel with a displacement of 32,000 tonnes and capacity of 7000 cars is expected to reduce emissions by 90 percent on their main trading routes. Other studies were performed in parallel worldwide, including kites/skysails as a solution. From an energy efficiency perspective, these solutions prevail in almost all conditions compared to other sources of energy as transmission losses are reduced, as shown below.

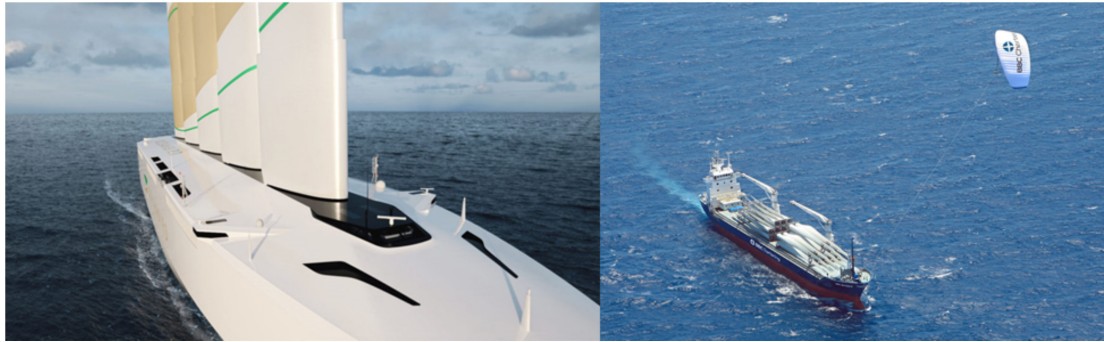

**Figure 9.** Oceanbird and Skysails as wind-powered solutions, Image Courtesy: Wallenius Marine and Skysails.

### 6.2. Economy of Scale and Increased Loading

Ships have grown in all dimensions to increase cargo capacity. Lately, mostly beam and length have increased, as draught restrictions in ports and fairways are a limiting factor. The growth of container vessels from the seventies onwards is illustrated in the Figure 10 below. This increase in size results in a better loading capacity and less required power per cargo. This is illustrated again based on the container vessels above. As the displacement and loading capacity do not increase linearly with length, beam or draught, increasing ship size is an effective way to increase ship efficiency. Making it possible to load as much as possible onto each ship hull will have a significant impact on power consumed per ton per nautical mile on a specific voyage.

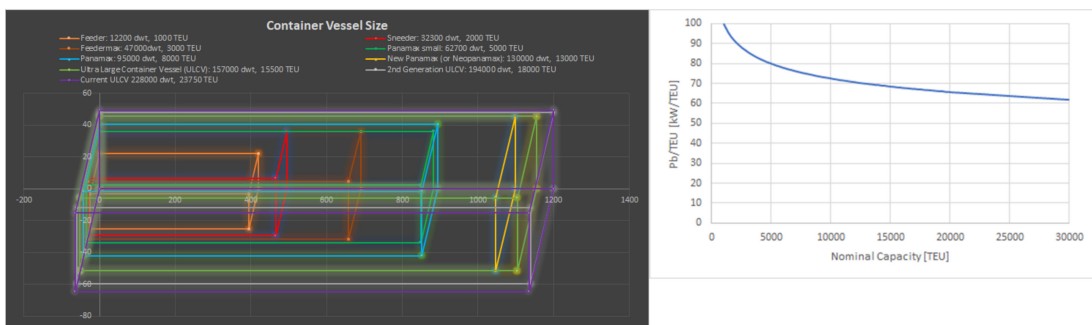

**Figure 10.** LEFT: Growth of container vessels size vs main dimensions, RIGHT: Growth in container-ship sizes and propulsion requirements per TEU, adapted from [35].

### 6.3. Hydrodynamic Properties, Power Management and Propulsion

Hydrodynamic properties, including ship resistance, power management and propulsion efficiency are significantly influencing energy efficiency as shown in Figure 10. Most of these influences are shown in the Figure 11 below. Historical reviews have shown that drivers for increased work and efficiency have been related to an increase in fuel costs, freight rates, and competition [36]. The study shows that, in general, the design efficiency of new ships improved significantly in the 1980s, was at its best in the 1990s, and deteriorated after that. The design efficiency for bulkers in the 1980s and 1990s was up to 10% better than in the period 1999–2008. A similar pattern can be observed for tankers and container ships as shown in Figure 12.

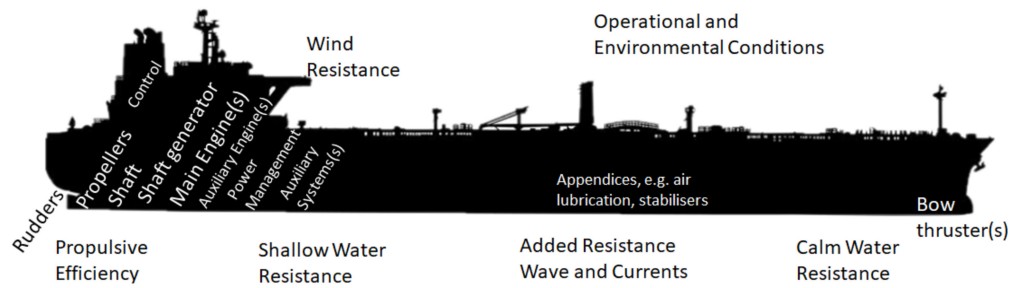

**Figure 11.** External and internal factors influencing the energy efficiency of ships hydrodynamics and propulsion system.

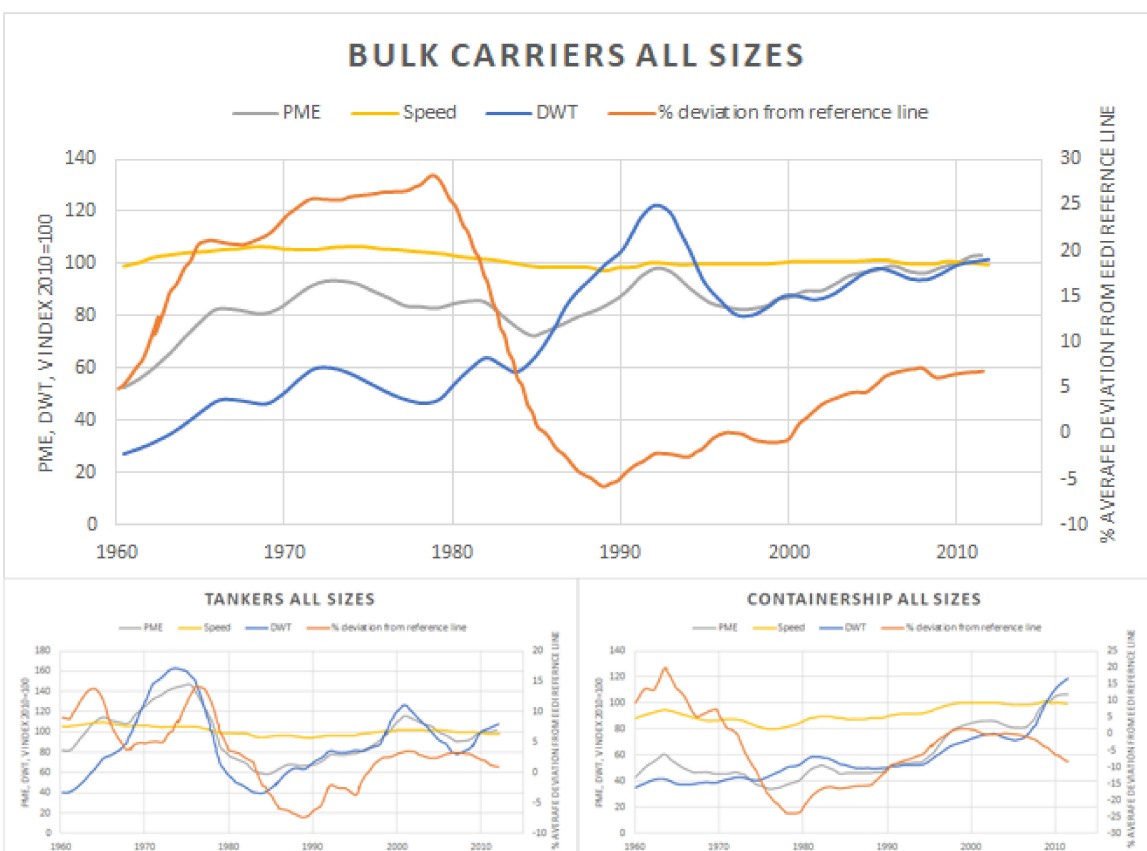

**Figure 12.** Development of design efficiency, main engine power, speed and capacity of new bulk carriers, 1960–2012, Design efficiency is defined as the EIV divided by EEDI reference line, averaged across all ships built in certain years. DOWN: Development of design efficiency, main engine power, speed and capacity of new tankers and container ships up to 2012, adapted from [36].

A historical analysis in the study [36] concluded that large improvements in design efficiency occurred within a relatively short period of time. Depending on the ship type, the average efficiency improved by 22–28% within a decade. These improvements were purely market driven by a combination of sharply increasing fuel prices as well as constant or low freight rates.

### 6.4. Weight Reduction

Weight reduction in ship designs resulting in a lighter ship meant more payload, less fuel consumption, and fewer $CO_2$ emissions. Traditionally, lighter materials have been used in cruise vessels to increase or ensure ship stability. The benefits of lighter material often comes with different characteristics and behaviour of the material as well as insulation

capabilities or strength. Very few figures have been published on the detailed benefits in weight reduction on ship design.

### 6.5. HVA/C and Waste Heat Recovery

Especially on cruise and passenger vessels, but even on RoRo and PCC (Pure Car Carrier) vessels, a significant amount of energy is consumed for heating and ventilation. An example from Birka Cruises is shown below. The optimization of the HVAC system and waste heat recovery are examples of more energy-efficient solutions.

Wallenius has worked on auxiliary machinery use management in ports related to HVAC systems, increasing energy efficiency during the port stay [37].

Energy recovery ventilation (ERV) has been installed on Wallenius pure car carriers (PCC) engine rooms. It is the energy recovery process in HVAC systems that exchanges the energy in the normal exhaust air of a building or conditioned space, using it to treat (precondition) the incoming outdoor ventilation air.

During the warmer seasons, an ERV system pre-cools and dehumidifies, and during cooler seasons the system humidifies and pre-heats. An ERV system helps HVAC design meet higher energy efficiency demands. The passenger vessel *Viking Grace* was equipped with a waste-heat recovery system. The energy recovery system converts waste heat from the vessel's engines into electricity. Basically, the engines need cooling water so that they do not overheat and seize up. When the cooling water passes through the engines, it becomes hotter and produces heat. Usually, this waste heat is conducted out of the vessel resulting in energy lost, but here it is used to produce electricity. The potential for waste heat recovery on the vessel is to save up to 750 tons of fuel by providing up to 40% of the electricity needed for the passenger functions on board, which adds up to 1900 tons of $CO_2$ savings per year.

### 6.6. Installed Power and Decreased Design and Operational Speed

Based on the business case and operational requirements, a design speed for a ship is set. The specific transport energy demand is shown in Figure 13. Based on ship design, a certain ship resistance and propulsion efficiency is derived resulting in the required propulsion power. The typical range of energy demand for ocean going vessels compared to other means of transport is shown in the figure below [34].

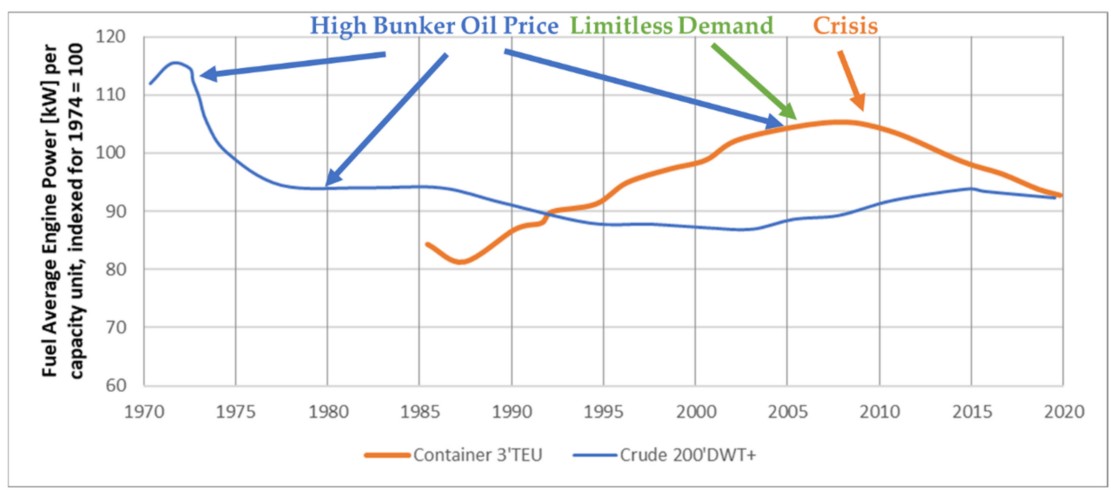

**Figure 13.** The development of installed energy power/capacity unit reflects the different stages shipping demand has been through, adapted from [1].

Ship speed has traditionally increased and decreased due to financial constraints, increased competition, variations in fuel price and low freight rates. In Table 1, two generations of propulsion power due to reduced design speed are shown [35].

**Table 1.** Effect of reduced design speed on ship efficiency, −7% due to economy of scale, −33% due to speed, adapted from [34].

|  | E-Class | Triple E-Class | Change (%) |
|---|---|---|---|
| Year Built | 2006 | 2013 |  |
| Capacity (TEU) | 14,770 | 18,270 | +24% |
| Full Displacement (t) | 208,000 | 258,000 | +24% |
| Deadweight (t) | 156,907 | 194,000 | +24% |
| Propulsion Power (kW) | 80,080 | 59,360 | −26% |
| Top Speed (knots) | 26 | 23 | −12% |
| Specific Propulsion Power | 5.42 | 3.25 | −40% |

## 7. Benchmark Performance

*KPIs and General Benchmarking*

Different ship and cargo types imply that measures differ on how energy efficiency should be considered. Shipping lines have made use of Key Performance Indicators (KPIs) for their sustainability reports, but no coherent measure has been established. Typical KPIs used for shipping efficiency are found to be:

- Fuel consumption, typically measured per hour/nautical mile, nautical mile and ton/nautical mile, and passenger];
- Fuel consumption of main engine(s), auxiliary engine(s), boilers;
- EEDI—Energy Efficiency Design Index (g $CO_2$/ton/mile)
- MRV—Total fuel consumption (m tonnes)/On laden (m tonnes)/total $CO_2$ emissions (m tonnes), possibly connected to transport work by annual average fuel consumption per distance (kg/n mile)/transport work (mass) (g/m tonnes·n miles)/(volume) (g/m$^3$·n miles)/(dwt) (g/dwt carried·n miles)/(pax) (g/pax·n miles)/(freight) (g/m tonnes·n miles) or by emissions annual average $CO_2$ emissions per distance (kg $CO_2$/n mile)/transport work (mass) (g $CO_2$/m tonnes·n miles)/(volume) (g $CO_2$/m$^3$·n miles)/(dwt) (g $CO_2$/dwt carried·n miles)/(pax) (g $CO_2$/pax·n miles)/(freight) (g $CO_2$/m tonnes·n miles)

It is for instance important that on passenger vessels the KPI's for the ships' carbon dioxide emissions per passenger include also, in addition to the propulsion of the ship, the heating and cooling of the ship, production of hot water and all electrical energy needed for passenger service, such as restaurants and hotel functions.

Looking at the Viking Line sustainability reports, the numbers from 2018 to 2019 differ significantly (Table 2), which shows the difficulty of finding good measurement standards. Similar tendencies can be seen in the Stena Lines sustainability report as shown in Figure 14 below [38].

Based on the analysis done on MRV and EEDI data, these statistics do not seem to be specific enough to make competent policies on how to reduce GHG emissions or on how ship owners can increase energy efficiency. Therefore, more detailed data collection and analysis methods are needed for ships to become more energy efficient.

**Table 2.** Carbon emissions on Viking Lines routes during 2018 and 2019, based on public data from [39,40].

| Carbon Emissions Average Value | 2018 kg per Passenger | 2019 kg per Passenger | 2018 kg per ton Freight | 2019 kg per ton Freight |
|---|---|---|---|---|
| Åbo-Långnäs | 24 | 6 | 4 | 21 |
| Stockholm-Långnäs | 33 | 9 | 6 | 29 |
| Åbo-Mariehamn | 32 | 8 | 5 | 28 |
| Stockholm-Mariehamn | 27 | 11 | 5 | 27 |
| Helsingfors-Mariehamn | 59 | 15 | 10 | 70 |
| Helsingfors-Tallinn | 18 | 7 | 2 | 19 |
| Mariehamn-Kapellskär | 27 | 23 | 13 | 53 |

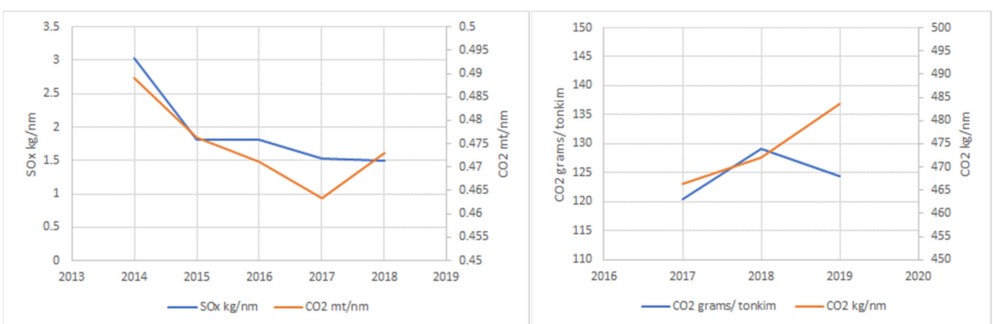

**Figure 14.** Sustainability Report Stena Line, Emission summary for 2018–2019 and 2019–2020, based on data from [38].

## 8. Review and Calibrate

Measures used for improving energy efficiency need to be reviewed and the consequences of the implementation need to be analysed. A known rebound effect for energy efficiency is the Jevron paradox, which that has been found in various transport applications.

*Jevon Paradox on Energy Efficiency—Rebound Effect*

In research there is quite a debate ongoing on what energy efficiency is needed and what the effect of energy efficiency implies in the long run.

The Jevons Paradox states that—in a longer perspective—an increase in efficiency in resource use will generate an increase in resource consumption rather than an expected decrease. Understanding the nature of the Jevons Paradox is therefore important in relation to any sustainability targets because it challenges the narratives behind sustainable energy policies striving for improvements in energy efficiency. The Jevons Paradox has generated an intense debate in the field of sustainability science not only among scientists attempting to prove or disprove its validity, but also in non-scientific communities [41–43].

The rebound effect (or take-back effect) is the reduction in expected gains from new technologies that increase the efficiency of resource usage because of behavioural, financial or other systemic responses. These responses usually tend to offset the beneficial effects of the new technology or other measures taken, as a reduction in the price for using a resource, which leads to increased usage.

## 9. Discussion and Conclusions

Based on the existing rules and regulations as well as the international and national targets, the way towards greener and fossil fuel-free shipping is given. Even though quite a few of the rules are not effective, it is clear that there is a need for technical and operational changes onboard vessels. These need to go together with policy changes to allow first movers to make a change and move the whole industry.

Based on the evaluation of technical solutions, it is logical that despite the choice of fuel or energy storage, it is essential to save energy and be more energy efficient than before. Improved energy efficiency of maritime transport can be achieved by

(1) Raising fill grades per ton and distance transported, achieved by

    a. Increased ship size

    b. Increased loading of vessels in loaded conditions; decreased loading in ballast conditions

    c. Optimised vessel loading

(2) Increasing the energy efficiency of freight movement per ton and distance transported, achieved by

    a. Operational measures

    b. Technical measures

    c. Organisational measures

(3) Reducing energy consumed per hour transported, achieved by, for example:

    a.    Operational measures (operating machinery and equipment more energy efficiently)

    b.    Technical measures (tank heating, HVAC)

    c.    Organisational measures such as implementing policies on how to run systems

(4) Reducing demand for freight transport, achieved by for example:

    a.    Optimised logistical chains and intermodal transport

    b.    Freight to lower energy-demand transport modes

Decarbonisation in addition might be achieved by reducing the carbon content of freight transport energy and shifting freight to lower-carbon transport modes where there is a high potential for shipping to provide alternatives.

There is high potential to make use of different solutions to increase energy efficiency. Many cost-efficient solutions relate to the operator where carrot and stick incentives can support the minimization of energy consumption, all from voyage planning with weather routing eco-driving, bonus-to-torque limitations, and limitation in company policies.

To make a difference in the climate impact on shipping and on reducing energy consumption onboard ship, it is not enough to collect high-level data such as MRV and work with spreading and calculation methods; more details are needed. Statistics do not seem to be specific enough to make competent policies on how to reduce GHG emissions and for ship owners to know how to increase energy efficiency. Therefore, more detailed data collection and analysis methods are needed for ships to become more energy efficient.

There are various systems available for data capture, storage and analysis. Certain of them are coupled to huge marine suppliers while others are stand-alones. Gaps and uncertainties identified in this study contain amongst others:

- Difficulty measuring sea currents in an exact way to measure or forecast sea and river currents
- Difficulties or high costs in equipping older vessels with suitable measurement devices or catch analogue signals. Certain engine suppliers have built their business cases around giving access to engine control units for providing such data.
- Non-inclusion of signals in standard coverage of data collection, such as rudder angle, which seems to be standard, as the purpose of data collection most of the time is only related to the engine, propulsion, wind and position.
- Availability of sensors and sensor systems that can be used in EX-classed areas on, for example, tankers.
- Suitable methods to easily assess and analyse data to find causal factors for energy consumption and capture potential to save energy onboard.
- Lack of standardization in data capturing, transmission and analysis.
- Cost drivers for systems are the retrofit installation costs for ships in service and to integrate individual technology components and to ensure that they perform robustly in combination.

Based on the work performed, the operator importance for onboard energy consumption was derived. A systematic improvement work is required in shipping companies to achieve energy savings. The regulations and policies as they are designed are not targeting reduced climate emissions and energy efficiency and thus need to be changed. When energy savings are achieved, it is due to customer requirements, cost pressure or individual driving forces in the companies. Data collection will lead to making better decisions in the lifecycle of the ship from knowledge-driven design to operation, redesign and lifetime extension. The potential brought up in the interviews showed energy savings of up to 35% on specific routes and up to 60% in specific maneuvers. This was made feasible by involving operators and crews in the decision-making process.

**Author Contributions:** Conceptualization, J.H.; methodology, J.H.; software, M.J.; formal analysis, J.H.; writing—original draft preparation, J.H.; writing—review and editing, M.J.; final adjustments, J.H.; project administration, M.J.; funding acquisition, J.H. All authors have read and agreed to the published version of the manuscript.

**Funding:** This research was funded by the Swedish Energy Agency/Energimyndigheten grant number 49301-1.

**Institutional Review Board Statement:** Ethical review and approval were waived for this study, due to the fact that interviewees were participating voluntarily and anonymous.

**Informed Consent Statement:** Informed consent was obtained from all subjects involved in the study.

**Data Availability Statement:** 3rd Party Data Restrictions apply to the availability of these data. Data was obtained from third parties. Interview transcripts can be shared on request.

**Conflicts of Interest:** The authors declare no conflict of interest. The funders had no role in the design of the study; in the collection, analyses, or interpretation of data; in the writing of the manuscript, or in the decision to publish the results.

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
