# Peer review of "State-of-the-Art Methods to Improve Energy Efficiency of Ships"

_jmse, doi:10.3390/jmse9040447_

Round 1
Reviewer 1 Report (Previous Reviewer 1)
From the title and the abstract from the paper, it says this paper is regarding to the state-of-the-art methods to improve energy efficiency of ships. However, the contents are mainly focus on the necessary of improvement of energy efficiency of ships and also the regulations of running ships.
1.The authors have stated that to make shipping more energy efficient is to collect more detailed data and new analysis methods. As a reviewer, I was expecting in the paper there should be more details about the methods which can improve energy efficiency of ships. However, I didn't find much about this. Maybe the authors should focus more on these parts since the main topic of this paper is about the methods.
2. the order of each chapters are confusing, for example, the regulations of current running ships should be put in the chapter of introduction. Please correct me if I am wrong.
3. Abbreviations should be stated when the first time using in the paper.
4. corresponding references should be cited when using the pictures or data from other researchers.
Author Response
From the title and the abstract from the paper, it says this paper is regarding to the state-of-the-art methods to improve energy efficiency of ships. However, the contents are mainly focus on the necessary of improvement of energy efficiency of ships and also the regulations of running ships.
1.The authors have stated that to make shipping more energy efficient is to collect more detailed data and new analysis methods. As a reviewer, I was expecting in the paper there should be more details about the methods which can improve energy efficiency of ships. However, I didn't find much about this. Maybe the authors should focus more on these parts since the main topic of this paper is about the methods.
Reply: There has been quite a bit of research on energy efficiency technical solutions. This paper focusses on the methods/ approach on how to implement changes in the organisations. There has been hardly any research on this in the maritime industry. Therefore, a general approach/ methodology is suggested and described in its detailed steps.
2. the order of each chapters are confusing, for example, the regulations of current running ships should be put in the chapter of introduction. Please correct me if I am wrong.
Reply: The focus is to describe the improvement cycle and there the drivers for change. I understand your intuition, but I wanted to keep the structure related to the improvement cycle.
3. Abbreviations should be stated when the first time using in the paper.
Reply: Added throughout the paper
- corresponding references should be cited when using the pictures or data from other researchers.
Reply: Already citations stated for all pictures from other researchers/ sources.
Reviewer 2 Report (Previous Reviewer 2)
(According to the reviewer's request, it will not be shown.)
Reviewer 3 Report (New Reviewer)
The article describes an important issue, but the structure of the article needs to be improved. There is a lot of abbreviations in the article, which are not written out the first time they are introduced fx. l.53 MRV, l. 127 HVAC etc.
The introduction is too short, and do not efficiently describe the problem and what is already known in the literature. The aim of the study is not transparent enough. What exactly is the aim of the study - need for detailed data collection and analysis? It is unclear what the focus is, is it on data, is it on awareness of seafarers or is it something else? In the conclusion some technical solutions are given, which confuses what the article is about.
The chapter Materials and Methods is rather confusing. I have difficulties to understand sentences in the lines 62-64 and the sentence in the l.60 is rather confusing. The first part of this chapter is rather background information and too some degree state of art, which could be move to the introduction. There are some issues with references. Reference 7 is minutes of meeting kick off – that’s not very scientific reference. There is also lack of broader literature what is written about energy efficiency. There are some articles in this area and there are not represented in the article. Be careful with repetition -the l. 92-95 is repetition from the introduction.
In the paragraph “Methods used”, the description of data is very limited, how many interviews were conducted and when and with how many respondents? Which literature were used and how the literature was searched for? Which methods were used? How the analysis was conducted?
In the next chapter “identification” it is unclear why the presented issues were selected, based on which analysis? L. 261 – language is more everyday language then scientific – “it gets obvious”
The description of the figure 3 is quite confusing- It is difficult to see that the data has a different format.
It is unclear what is the aim of the chapter “measure performance”. How does this chapter contribute to the study?
In the chapter “detailed data collection” it is not transparent where the results came from and which data were used. This relates to lacking description of the data in the method chapter.
How the next chapter “Identify relevant improvement area, approach or strategy” is connected to aim of the study about detailed data?
The main problem with this study is the lack of transparent aim and research problem. This is the reason why it is difficult for reader to follow the logic of the article. There are a lot of issues described in the study, which can be confusing. The focus is too some degree on technical solutions, which is not necessary connected to what was mentioned in the beginning of the study. There is no clear answer what kind of data is needed and for which purpose. The article will benefit to narrow the focus and “keep” on track and do not include too many perspectives. There is also a need to include more scientific articles about energy efficiency and better use of the ones, which are already included.
Author Response
The article describes an important issue, but the structure of the article needs to be improved. There is a lot of abbreviations in the article, which are not written out the first time they are introduced fx. l.53 MRV, l. 127 HVAC etc.
Reply: Added
The introduction is too short, and do not efficiently describe the problem and what is already known in the literature. The aim of the study is not transparent enough. What exactly is the aim of the study - need for detailed data collection and analysis? It is unclear what the focus is, is it on data, is it on awareness of seafarers or is it something else? In the conclusion some technical solutions are given, which confuses what the article is about.
Reply: Changed, moved some parts as suggested by another reviewer and some more text on the objectives added,
The chapter Materials and Methods is rather confusing. I have difficulties to understand sentences in the lines 62-64 and the sentence in the l.60 is rather confusing. The first part of this chapter is rather background information and too some degree state of art, which could be move to the introduction.
Reply: Changed
There are some issues with references. Reference 7 is minutes of meeting kick off – that’s not very scientific reference. There is also lack of broader literature what is written about energy efficiency.
Reply: No relevant scientific references are found for this specific topic, therefore grey literature has been used as well.
There are some articles in this area and there are not represented in the article. Be careful with repetition -the l. 92-95 is repetition from the introduction.
In the paragraph “Methods used”, the description of data is very limited, how many interviews were conducted and when and with how many respondents? Which literature were used and how the literature was searched for? Which methods were used? How the analysis was conducted?
Reply: Information added
In the next chapter “identification” it is unclear why the presented issues were selected, based on which analysis? L. 261 – language is more everyday language then scientific – “it gets obvious”
Reply: Information added, and language adjusted.
The description of the figure 3 is quite confusing- It is difficult to see that the data has a different format.
Reply: Changed
It is unclear what is the aim of the chapter “measure performance”. How does this chapter contribute to the study?
Reply: This is part of the cycle described above. If you want to make changes, the literature and interviewees states that this is an important part of the work on reduced energy consumption.
In the chapter “detailed data collection” it is not transparent where the results came from and which data were used. This relates to lacking description of the data in the method chapter.
Reply: It is stated that it is based on the interviews.
How the next chapter “Identify relevant improvement area, approach or strategy” is connected to aim of the study about detailed data?
Reply: In the improvement cycle, this step is used as decision support for implementing technical measures.
The main problem with this study is the lack of transparent aim and research problem. This is the reason why it is difficult for reader to follow the logic of the article. There are a lot of issues described in the study, which can be confusing. The focus is too some degree on technical solutions, which is not necessary connected to what was mentioned in the beginning of the study. There is no clear answer what kind of data is needed and for which purpose. The article will benefit to narrow the focus and “keep” on track and do not include too many perspectives. There is also a need to include more scientific articles about energy efficiency and better use of the ones, which are already included.
Reply: We have tried to make this clearer in the update. In order to make efficient decisions for increased energy efficiency, detailed data collection is needed. This needs to be made available to the crews and operators.
Round 2
Reviewer 1 Report (Previous Reviewer 1)
The paper has been revised and updated according to the all the advises given. Several points have been clarified.
Now the paper is well written and the topic is generally of interest for the journal readers. The paper deserves to be considered for the journal of marine science and engineering.
Reviewer 2 Report (Previous Reviewer 2)
(According to the reviewer's request, it will not be shown.)
Reviewer 3 Report (New Reviewer)
Thank you for rewritten version of the article. It improved it very much, it is much more structured and it is easy for reader to follow your argumentation.
I don't have any additional comments.
This manuscript is a resubmission of an earlier submission. The following is a list of the peer review reports and author responses from that submission.
Round 1
Reviewer 1 Report
The topic of the article is interesting, however, this article is so difficult to read, 1) from section 2 to section 4, the ordering of the the paper is totally mess up. The whole section 2 and also section 3 are so confusing. 2) all the abbreviations should be clear stated, from Abstract to Conclusion so many abbreviations are not clarified, therefore, it is difficult to read this paper. 3) Missing of citations, for example, in the Introduction section from line 82 to 72, no citations are included. 4) the answers to the questions (from line 117 to line 132) are also difficult to find. And these are essential to this paper.
Reviewer 2 Report
(According to the reviewer's request, it will not be shown.)